# Exact Online Learning with Gamma-memory delays for Accurate Feedforward SNNs

## Abstract

Spiking Neural Networks (SNNs) promise energy-efficient, low-latency AI through sparse, event-driven computation. Neuromorphic hardware can realize this efficiency by exploiting high temporal resolution, as precise spike timing supports compact and sparse information processing. Yet, training SNNs under fine temporal discretization remains a major challenge. In state-of-the-art approaches, spiking neurons are modeled as self-recurrent units, embedded into recurrent networks to maintain state, and trained with BPTT or RTRL variants based on surrogate gradients. We show that these methods scale poorly with temporal resolution, while online approximations methods are inherently unstable. We solve this problem by developing recursive memory structures combined with a linear–nonlinear interpretation of spike-train generation in spiking neurons: the SpikingGamma model. We show that SpikingGamma models support direct error backpropagation without surrogate gradients, can learn fine temporal patterns with minimal spiking in an online manner, and scale feedforward SNNs to complex tasks with competitive accuracy, all while being insensitive to the temporal precision of the model. Our approach offers both an alternative to current recurrent SNNs trained with surrogate gradients, and a direct route for mapping SNNs to neuromorphic hardware.

## 1 Introduction

Inspired by the brain, Spiking Neural Networks (SNNs) (Maass, 1997) hold promise for energy-efficient AI models (Li et al., 2023; Dampfhoffer et al., 2022) as they use highly sparse and discrete spikes for communication, and enable event-driven, asynchronous information processing. This allows neurons to respond only when needed, achieving low-latency responsiveness while lowering energy consumption, especially when tasks exhibit extended temporal dynamics. Consequently, SNNs are well-suited for real-world tasks such as autonomous driving (Martínez et al., 2024), auditory signal processing (Baek & Lee, 2024), and drone control (Hagenaars et al., 2020).

For small scale tasks, current SNNs now demonstrate competitive performance with classical neural networks (Hammouamri et al., 2023; Eshraghian et al., 2023). Progress here was built on treating spiking neurons as self-recurrent neural units, and using a so-called Surrogate Gradient (SG) to backpropagate errors through the discontinuity of the spiking process (Neftci et al., 2019). Yet, as SNNs are intrinsically treated as time-stepped RNNs, they inherit many of the drawbacks of classical RNNs, including vanishing and exploding gradients, and the need to maintain or approximate past influences on current state to train these networks with algorithms like BPTT and RTRL (Zenke & Vogels, 2021). As these approaches incur prohibitive memory and/or timestep complexity, online approximations to these algorithms have been developed (Kaiser et al., 2020; Wang et al., 2024), however, being approximations, they typically fail when scaling either through time (sequence length) or space (network size), where the approximate nature of SGs compound these issues. In theory, SGs also intrinsically limit the degree of sparseness in a networks, as excessive sparsity amplifies gradient vanishing, collapsing training at a critical transition point (Zenke & Vogels, 2021).

At the same time, the deployment of SNNs in efficient hardware requires a careful correspondence between model network dynamics and actual dynamics of the physical substrates (Cramer et al., 2022; Ko et al., 2024; Koopman et al., 2025). Modeling actual physical dynamics however is usually achieved via fine-grained temporal simulation, resulting in the need to use many small timesteps when training a network (Ko et al., 2024). This requirement however conflicts with the current SNN

training paradigm, also, as we show, for online approximations to BPTT/RTRL like FPTT (Kag & Saligrama, 2021) and ES-D-RTRL (Wang et al., 2024).

Here, we introduce the SpikingGamma model and demonstrate how it resolves this issue. The model builds on earlier work including the Gamma-model (De Vries & Principe, 1992), the Temporal Kernel RNN (Sutskever & Hinton, 2010) and the Fractionally Predictive SNN (Bohte, 2011; Rombouts & Bohte, 2010). The SpikingGamma model employs adaptive recursive memory to efficiently create an increasingly smoothed delayed representation of past inputs, where, as in (Yoon, 2016; Rombouts & Bohte, 2010), sigma-delta spike-coding encodes the rectified internal state of a neuron into a spike-train which is then effectively decoded at the received neuron, similar to pulsed sigma-delta coding in electrical circuits (Yoon, 2016). Combining adaptive recursive memory with sigma-delta spike-coding removes self-recurrency, and enables feedforward SNNs to be trained directly with error-backpropagation without the use of SGs, while being able to learn useful temporally disjunct feature patters. Notably, thus formulated the networks can be trained with arbitrary temporal precision.

In this model of computation, the smoothed delays the in the SpikingGamma model enables neurons to detect the co-occurrence of temporally disjunct features, without externally maintaining memory through for example persistent activity. Discrete delays have been demonstrated to be highly effective for achieving high-performance in SNNs (Hammouamri et al., 2023). Ludvig et al. (2008) also already demonstrated that such a smoothed-delay model can account for peculiarities of delayed-reward reinforcement learning in biology, where the prediction error response to a reward stimulus is directly suppressed when a reliable preceding cue-stimulus is learned. In their model, a stimulus generates a multitude of differentially delayed microstimulia affecting downstream neurons, enabling reinforcement learning to directly connect temporally disjunct stimuli and rewards.

Here, we show how SpikingGamma SNNs compete with current state-of-the-art SNN approaches, can learn very sparse and precise temporal coding over long timespans, while being insensitive to the temporal resolution of the simulated dynamics. Our approach thus presents an alternative to current recurrent SNNs based on Surrogate Gradient training, where SpikingGamma SNNs can be trained and computed online and at arbitrarily high temporal resolution.

## 2 RELATED WORK

Where surrogate gradients have enabled training of SNNs with standard recurrent deep learning approaches, several alternatives to BPTT and RTRL have been explored to improve the training efficiency of SNNs. ANN-to-SNN conversion, while straightforward and scalable, fails to exploit the temporal dynamics and event-driven sparsity of SNNs, and most existing methods remain constrained to inefficient rate-based coding schemes (Zhou et al., 2024). Various approximations of RTRL and BPTT have been proposed that maintain temporal traces for credit assignment or regularize gradients through time rather than computing exact gradients, such as DECOLLE (Kaiser et al., 2020), OSTL (Bohnstingl et al., 2022), OTPE (Summe et al., 2024), FPTT (Yin et al., 2023), OTTT (Xiao et al., 2022), and ES-D-RTRL (Wang et al., 2024). These methods have shown limited success on tasks demanding complex temporal reasoning. Another line of work seeks to exploit the inherent sparsity of SNN activity to reduce the memory footprint of BPTT, such as EventProp (Mészáros et al., 2025), though its scalability to large models and datasets remains unexplored.

Our work is inspired by a different idea: maintaining a compressed representation of history within the forward pass to enable temporal learning without full backpropagation through time. Related ideas have been explored in ANNs. For example, in the Temporal Kernel RNN (Sutskever & Hinton, 2010) all neurons are linked through all timesteps using a learned weight matrix, akin to delays, and WaveNet (Van Den Oord et al., 2016) uses dilated causal convolutions to model long-range temporal dependencies within a feedforward architecture, with performance comparable to LSTM-based RNNs. The Gamma Model (De Vries & Principe, 1992) similarly maintains adaptive internal memory to represent past inputs, functioning like a form of temporal convolution. Notably, so-called time-cells have been identified in the brain with similar delayed response properties (Liu et al., 2019), and Gamma-models have been proposed to account for internal dynamics in biological neurons such as spike-rate adaptation (Drew & Abbott, 2006). A related approach was applied to SNNs to avoid BPTT (Bohte, 2011), but the delay-structure there was incompatible with scalable learning as it could not be computed in a recurrent manner – the concepts introduced in (De Vries & Principe, 1992) and (Bohte, 2011) however form the basis for our work.

## 3 METHODS

### 3.1 THE NEURON MODEL

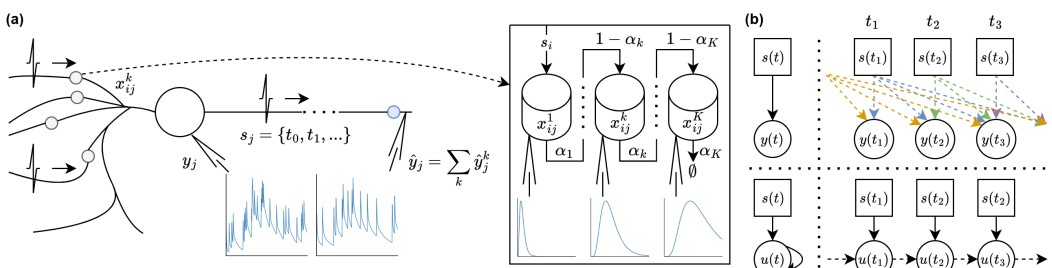

Figure 1: **(a)** Neural processing model: input spikes enter the network from the left. At the synapses, spikes generate weighted currents that evolve over multiple timescales, where currents are implemented as a cascade of leaky "buckets" that drain into one another at different rates ($\alpha_k$) (block on the right). Within the neuron, the resulting synaptic responses are weighted according to their timescales and summed to produce a continuous neuronal signal ($y_j$). This signal is then converted back into spikes, allowing downstream synapses to reconstruct an estimate of the original signal ($\hat{y}_j$). **(b)** Transition of neuron states over time in SpikingGamma (top) versus recurrent architectures (bottom). While the simplified step view (left) shows both behaving similarly, the time-dependency view (right) reveals a clear difference: SpikingGamma has access to the entire history via buckets for each timestep, whereas recurrent models rely on internal states (e.g., membrane potential $u$) and therefore require BPTT for training.

In the SpikingGamma model, we base the spiking neuron model on a Linear-Non-Linear Sigma Delta (LNL-SD) temporal filtering model (Rombouts & Bohte, 2010), where a neuron $j$ computes an internal signal $y_j$ as the sum of the weighted and filtered inputs into the neuron. To implement sigma-delta spike-coding, the neuron additionally tracks a signal $\hat{y}_j$ as the sum of spike-triggered refractory responses (Gerstner & Kistler, 2002): similar to sigma-delta coding, this sum of refractory responses approximates the rectified $y_j$ by emitting a spike and adding a refractory response whenever the positive approximation error between $\hat{y}_j$ and $y_j$ exceeds a threshold $\vartheta$ (Zambrano et al., 2019). The approximation of $y_j$ is thus encoded by the spike-train determined by those threshold exceedances as downstream neurons can reconstruct the signal $\hat{y}_j$ at their input – the neural processing model is illustrated in Figure 1. Mathematically, the signal $y_j$ is computed as:

$$y_j(t) = \text{ReLU}(x_j(t)) \tag{1}$$

with $x_j$ being the unrectified neuron signal. This is computed by filtering the synaptic input signals $x_{ij}^k$ from neuron $i$ to neuron $j$ for $k < K$ with each synapse having $K$ temporal kernels. The filters can be instantiated as per-neuron or per-synapse. For per-neuron this is described as:

$$x_j(t) = \sum_k \sum_i x_{ij}^k(t) \cdot v_j^k, \tag{2}$$

where $v_j^k$ is a parameter that weights the value of kernel $k$ for neuron $j$. We refer to this as the "bucket weight". The synaptic signal $x_{ij}^k$ is then computed as:

$$x_{ij}^k = \hat{y}_i^k(t) \cdot w_{ij}, \tag{3}$$

where $w_{ij}$ is the synaptic weight, and $\hat{y}_i^k$ is the temporal kernel $k$ that estimates the signal of the upstream neuron $i$, as encoded by output spikes $t_i$ from neuron $i$ to $j$:

$$\hat{y}_i^k(t) = \sum_{t_i < t} \kappa^k(t - t_i), \tag{4}$$

where $t_i$ denotes spike-times of neuron $i$ up to time $t$, and $\kappa^k$ being a set of temporal kernel functions implementing different delays.

Following the Gamma-model (De Vries & Principe, 1992; Drew & Abbott, 2006) the kernels $\kappa^k$ are computed as a series of "buckets" that spill over into each other:

$$\hat{y}_i^k(t) = \begin{cases} \hat{y}_i^k(t-1) \cdot \alpha_k + s_i(t) \cdot 2 \cdot \vartheta_i(t), & \text{if } k = 0 \\ \hat{y}_i^k(t-1) \cdot \alpha_k + \hat{y}_i^{k-1}(t-1) \cdot (1-\alpha_k), & \text{if } k > 0 \end{cases} \tag{5}$$

with $\alpha_k$ specifying the transfer rate between buckets $k$ and $k-1$, $\vartheta_i(t)$ being the threshold function of the spiking neuron, as defined in Section 3.3, and $s_i(t) = 1$ if $t \in \{t_i\}$ for an output spike-train $\{t_i\}$, else 0. As this is a set of linearly coupled differential equations, each $\hat{y}_i^k$ can be reformulated as Eq. (4) with each $\kappa^k$ not relying on any other kernel. Thus, each $\kappa^k$, and by extension each bucket $\hat{y}_i^k$ can be modeled in a feedforward fashion without recurrency – that is, knowing the spike times of the inputs $t_i < t$, one can exactly compute the value of $\hat{y}_i^k(t)$ for any $t$ (Bohte, 2011).

Spike thresholding implements sigma-delta spike-coding and is done by first computing a variable $z_j(t)$ that is similar to the membrane potential by subtracting an approximation of the signal $\hat{y}_j$, which is what will encoded by the to be emitted spikes, from the actual internal signal $y_j$:

$$z_j(t) = y_j(t) - \hat{y}_j(t), \tag{6}$$

with

$$\hat{y}_j(t) = \sum_k \hat{y}_j^k(t), \tag{7}$$

where $\hat{y}_j^k$ is defined per Eq. (5) for the output of neuron $j$ itself. When the difference exceeds a threshold, the signal approximation $\hat{y}_j(t)$ is then updated by emitting a new spike:

$$\begin{aligned} s_j(t) &= z_j(t) > \vartheta_j(t), \\ t_j &= t, \quad \text{if } s_j(t) = 1 \end{aligned} \tag{8}$$

with $\{t_j\}$ being the output spike-train. Figure 2 visualizes the complete forward path together with example signal values at each stage.

Both the signal approximation at the output of the neuron and the signal received at the downstream synapses are actually the same. Thus, it is possible to directly use $\hat{y}_j^k(t)$ for computations at those downstream synapses (i.e., for Eq. (3)). To filter per synapse, the only change need to the model above is to use individual per-synapse bucket weights $v_{ij}^k$ instead of $v_j^k$.

### 3.2 Error-backpropagation

For classification, given some desired output label $y_{\text{true}}(t)$ and actual neuron signal of the output neuron $y_{\text{out}}(t)$, we can compute a loss such as the Cross-Entropy (CE) loss $L(t) = L^{\text{CE}}(y_{\text{out}}(t), y_{\text{true}}(t))$. If the task requires a precise spike timing, we can instead use a Mean Squared Error (MSE) loss between the expected $\hat{y}_{\text{true}}$ that would follow from a spike in the correct class at the right time, and actual neuron output $\hat{y}_{\text{out}}$. Then, $L(t) = L^{\text{MSE}}(\hat{y}_{\text{out}}(t), \hat{y}_{\text{true}}(t))$.

As illustrated in Figure 2, for each discrete timestep $t$, we use the variables at that timestep to compute the loss with respect to the synaptic weights $w_{ij}$ and the bucket weights $v_j^k$ (or $v_{ij}^k$ in the case of per-synapse filtering) (we omit the explicit time index $(t)$):

$$\frac{\partial L}{\partial w_{ij}} = \frac{\partial L}{\partial \hat{y}_j} \frac{\partial \hat{y}_j}{\partial y_j} \frac{\partial y_j}{\partial x_j} \sum_k \frac{\partial x_j}{\partial x_{ij}^k} \frac{\partial x_{ij}^k}{\partial w_{ij}}, \tag{9}$$

$$\frac{\partial L}{\partial v_j^k} = \frac{\partial L}{\partial \hat{y}_j} \frac{\partial \hat{y}_j}{\partial y_j} \frac{\partial y_j}{\partial x_j} \frac{\partial x_j}{\partial v_j^k}, \tag{10}$$

where $\frac{\partial L}{\partial \hat{y}_j}$ follows directly from the definition of the loss function. To avoid having to backpropagate through spikes, we exploit the definition of $\hat{y}_j$ as being an approximation of $y_j$, thus having $\frac{\partial \hat{y}_j}{\partial y_j} = 1$.

The other terms follow directly from their forward pass, namely: $\frac{\partial y_j}{\partial x_j} = \text{ReLU}'(x_j)$, $\frac{\partial x_j}{\partial x_{ij}^k} = v_j^k$, and finally $\frac{\partial x_{ij}^k}{\partial w_{ij}} = \hat{y}_i^k$ and $\frac{\partial x_j}{\partial v_j^k} = \sum_i x_{ij}^k$ (again, replace $v_j^k$ with $v_{ij}^k$ for per synapse filtering). To traverse the computational tree over a layer, the last term of Eq. (9) is replaced with $\frac{\partial x_{ij}^k}{\partial \hat{y}_i} = w_{ij}$.

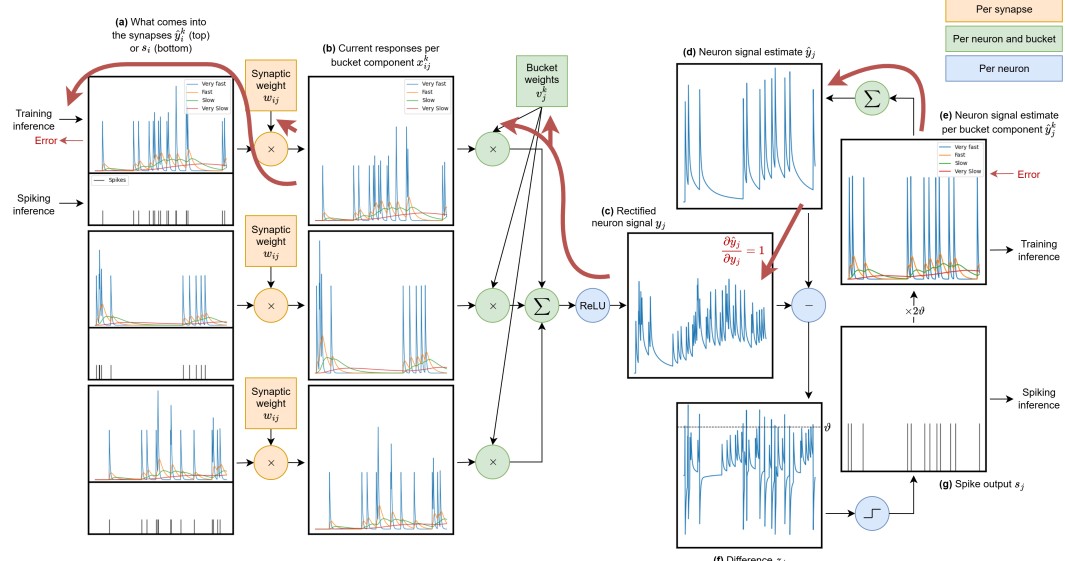

Figure 2: Signal evolution and error propagation in the neuron model. **(a)** At each input synapse, the neuron receives a signal. During training this is the signal estimate $\hat{y}_i^k$ of the presynaptic neuron, while during spiking inference (e.g., on a neuromorphic chip) this estimate is reconstructed from the incoming spike train $s_i$ following Eq. (5). **(b)** Each input is scaled by its synaptic weight $w_{ij}$, producing a current response per synapse and bucket. **(c)** These responses are weighted per bucket by the bucket weights $v_j^k$ (or $v_{ij}^k$ if weighted on synapse level), then accumulated across the buckets and finally rectified, forming the neuron signal $y_j$. **(d)** For discretizing this analog signal back into spikes, the neuron maintains a running estimate $\hat{y}_j$ that is encoded by the output spikes. **(e)** This estimate is expressed in the same kernel basis as the input, ensuring consistency across layers. Because $\hat{y}_j^k$ is mathematically identical to the input representation $\hat{y}_i^k$ (Eq. (5)), it can be passed directly to downstream synapses during training, without spike decoding. **(f)** Whenever the mismatch $z_j = y_j - \hat{y}_j$ exceeds a threshold, a correction is triggered. **(g)** This results in a spike output $s_j$, which is added back into the estimate. The red arrows indicate the error pathway during training: the error flows from the signal estimate back to the neuron signal and further to the inputs and parameters. Importantly, the error bypasses the spikes, eliminating the need for surrogate gradients.

## 3.3 INITIALIZATION, REGULARIZATION AND CONSTRAINTS

Before training, parameters are initialized. During training, we apply regularization and add constraints to prevent overfitting, improve model generalization, and increase sparsity. We use standard methods as summarized below, and additionally describe our *Adaptive Thresholding* and *Bucket-transfer Rate Initialization* in detail.

**Dropout on neuron signal**. Randomly zero out the neuron signal $y$ with a given probability during training. The decision to drop is independent according to a Bernoulli distribution. Applied to every layer except the output layer.

**Layer normalization**. Normalizes the neuron signal before rectification ($x$) based on layer statistics (Ba et al., 2016): $x_{\text{norm}} = \frac{x - \text{E}[x]}{\sqrt{\text{Var}[x] + \epsilon}} \cdot \gamma + \beta$, with $\gamma$ being the gain, $\beta$ the bias (both trainable), and $\text{E}[x]$ and $\text{Var}[x]$ respectively the mean and variance of the neuron signal over all neurons in the layer.

**Gain loss**. Penalizes the magnitude of the gain term $\gamma_l$ of the normalization layers: $L_{\text{gain}} = G \cdot \sum_l \gamma_l$, where $G$ is a constant that affects the relative importance of the gain loss term, and $l$ is the normalization layer index. This shrinks the signal, in turn reducing the number of spikes.

**Weight initialization**. Synaptic weights and biases on fully-connected and convolutional layers are sampled from $\mathcal{U}(-\sqrt{k}, \sqrt{k})$ with $k = \frac{1}{\text{input features}}$. The bucket weights are sampled from $\mathcal{N}(0, 0.1)$.

**Adaptive thresholding.** In the sigma-delta spike-coding model, whenever the difference between $y$ and $\hat{y}$ exceeds a fixed threshold, a constant input $s$ (pulse) is added to $\hat{y}$. As $y$ increases, the leak from the buckets (which grows relative to $\hat{y}$) eventually balances this constant input. Beyond that point, if $y$ reaches an even higher value, the leak will prevent $\hat{y}$ from approaching $y$. Consequently, the relation $\frac{\partial \hat{y}}{\partial y} = 1$ no longer holds, which can disrupt training.

To overcome this, we use an adaptive thresholding mechanism (Zambrano et al., 2019) that increases the threshold, and therefore the amount added to the buckets, in proportion to $\hat{y}(t)$:

$$\vartheta(t) = \vartheta_0 + \hat{y}(t) \cdot m_f \tag{11}$$

where $\vartheta_0$ is the minimum threshold, and $m_f$ is a constant scaling factor. In our experiments, we set $m_f = \vartheta_0$. With this adaptive thresholding, $\hat{y}$ can approximately track any positive value of $y$, as illustrated in Figure 7 in the Appendix. As the adaptive threshold can be calculated at the receiving neuron side, binary spikes can still be used, or graded spikes absent such downstream calculation (Zambrano et al., 2019).

**Bucket transfer rate initialization.** Following (Bohte, 2011), we ensure that the combined sum of buckets follows a power-law-like curve to facilitate learning long-range temporal filtering. To achieve this, we generate linearly separated values $l_k \in \text{linspace}(L_{\text{start}}, L_{\text{end}}, K)$, with $L_{\text{start}} \in (0, 1)$ being the starting value, $L_{\text{end}} \in (0, 1)$ being the final value (both included in the generated value range), and $K$ the number of values, one for each bucket. This is then used to compute the transfer rates,

$$\alpha_k = (l_k)^F, \tag{12}$$

with $F \in (0, 1)$ being the transfer rate factor. Combined with $L_{\text{start}}$, $L_{\text{end}}$, and $K$ this determines the shape of the curve. For all experiments, $L_{\text{start}} = 0.1$ and $L_{\text{end}} = 0.9$. In Table 5 in the Appendix, the neuron response for different values of $F$ are shown.

## 4 EXPERIMENTS AND RESULTS

We demonstrate how deep feedforward SNNs using SpikingGamma models can 1) learn to detect and respond to precisely timed spikes, with minimal spiking 2) achieve accuracy in temporally sensitive benchmarks competitive and exceeding current online learning approaches for SNNs, and 3) scale without loss of accuracy to fine-temporal resolutions which are infeasible with exact recurrent learning and where current approximate online learning approaches fail.

### 4.1 LEARNING EXACT TIMINGS THROUGH DELAYED RESPONSES

We demonstrate the effectiveness of our approach through two examples involving temporally structured computations.

**Learning Delays.** In the first task, a single input spike needs to be propagated with a precise delay through a hidden neuron to a single output neuron. This task allows us to compare BPTT learning with BP learning in the feedforward SpikingGamma model.

Specifically, an input is provided at $t = 0$, and the network is trained to produce an output at $t = 150$. Each of the two trainable layers in the network contains a single neuron, with only the bucket weights being trainable while the synaptic weights between input and hidden, and hidden and output neuron remain fixed. We compute the MSE loss between the actual output and expected output per timestep. We define output per the trace $\hat{y}$ that follows from the spike output.

For BPTT with SGs, the gradient is computed as:

$$\frac{\partial L}{\partial v_j^k} = \frac{\partial L}{\partial \hat{y}_j(t)} \frac{\partial \hat{y}_j(t)}{\partial s_j(t)} \frac{\partial s_j(t)}{\partial z_j(t)} \frac{\partial z_j(t)}{\partial y_j(t)} \frac{\partial y_j(t)}{\partial x_j(t)} \left( \sum_{t' < t} \frac{\partial x_j(t)}{\partial x_j(t')} \frac{\partial x_j(t')}{\partial v_j^k} + \frac{\partial x_j(t)}{\partial v_j^k} \right), \tag{13}$$

BPTT propagates back through the spikes using an SG $z_j$ and then back in time through the buckets, rather than treating the system as a purely feedforward network as in the SpikingGamma model.

In Figure 3, we plot for both methods the neuron signal dynamics after training (on the left), and how the bucket weights evolved during training (on the right). Both approaches succeed in learning the

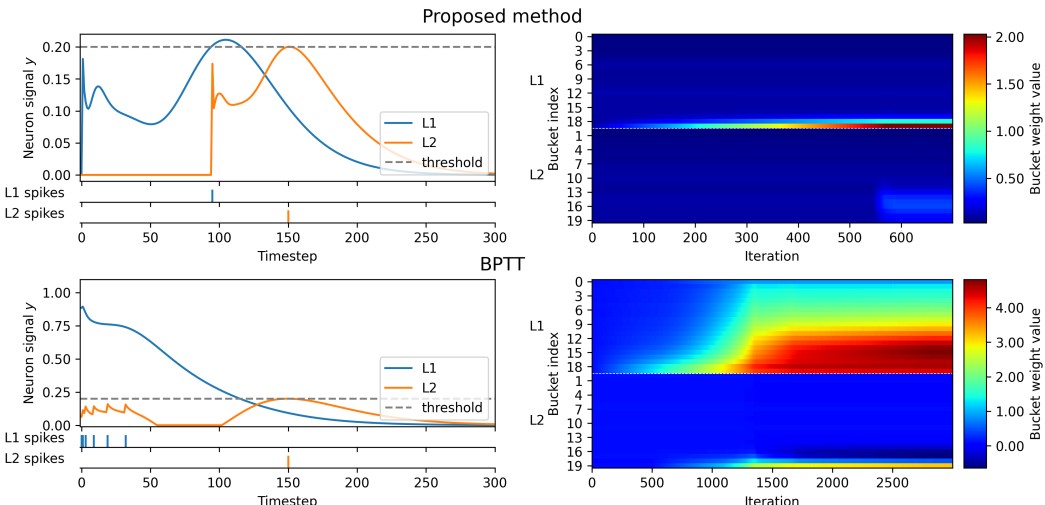

Figure 3: Comparison between SpikingGama (top) and BPTT with SGs (bottom). Left shows the neuron dynamics after converging, and right the bucket-weight evolution during training.

long delay. However, the SpikingGamma SNN can precisely target delay kernels at the timesteps where signal increases should occur, allowing it to realize the delay with the minimum number of spikes (two), whereas BPTT with SGs tends to reinforce a broader band of kernels, an effect that may reflect diffuse temporal credit assignment and the smooth surrogate derivative spreading gradients across nearby timesteps, consistent with findings in (Li et al., 2024).

**Learning Delayed Coincidence Detection.** In the second task, shown in Figure 4, the aim is to learn delayed coincidence detection. This task models a sound arriving from a particular direction as being represented by a distinct pair of spike times across two input channels, where the class is determined by the relative time-difference between the two spikes. This mimics the mechanism used by barn owls to localize prey based on subtle interaural time differences in the arrival time of spikes as a function of azimuth of the prey relative to the owl's head direction (Carr & Konishi, 1990).

The input consists of a spike pair (left (L) spike time, right (R) spike time) as defined by the class, plus jitter drawn from $\mathrm{Uniform}(0, 2)$. For the four classes, the pairs are: $(4, 60)$, $(4, 20)$, $(20, 4)$, and $(60, 4)$. The target for each class is a single output spike at time 200 from the corresponding class neuron, with other output neurons remaining silent (see Figure 4b). The loss is computed the same way as for learning delays.

Both the left and right input have their own unique synaptic connections to all four output neurons (so 8 connections in total), thus each output neuron has two synaptic inputs (see Figure 4a). Each of these synapses has 25 buckets, where the bucket weights are trainable and applied on synapse level. Within each neuron, the synaptic responses of both inputs are summed up.

Figure 4c shows the synaptic responses after training. Effectively, each synapse learns a specific approximate delay such that for a specific pair of input spike times, their sum (L+R) results in a spike for the right class at the right time (see Figure 4d). Accuracy (per time-to-first-spike coding) is 100% after training, with minimal spiking (maximal efficiency). This demonstration suggests that the SpikingGamma SNN is well suited for temporally structured classification tasks and learns in a way that aligns with biological neural coincidence detection.

## 4.2 SCALING THROUGH SPACE AND TIME

To demonstrate scaling to larger networks, problems and temporal precision, we evaluate the SpikingGamma model on three widely used benchmarking datasets. We use DVS Gesture (Amir et al., 2017), SHD, and SSC (Cramer et al., 2020); while SHD and SSC can be handled with relatively shallow fully connected network topologies, they remain challenging because of the need to capture

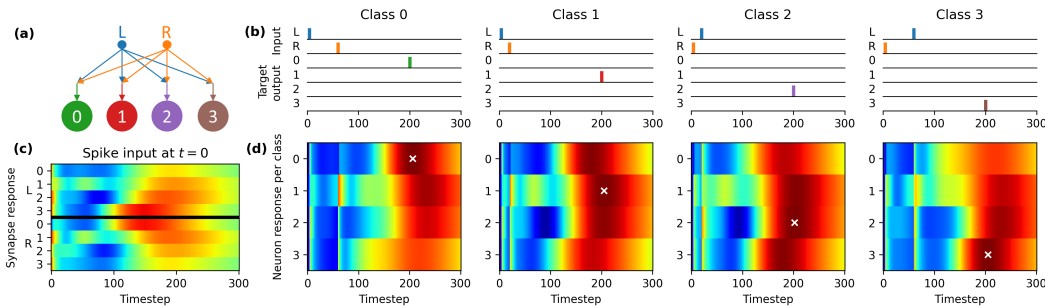

Figure 4: **(a)** The architecture of the coincidence detection network. **(b)** Input and target spikes. **(c)** The weighted bucket activation per synapse after training, given an input spike input at $t = 0$. **(d)** Output neuron response to the correct class input as given in (b) with the trained synaptic responses as in (c) summed within the output neuron. Output spike (white cross) is emitted at about $t = 200$ in the correct class.

rich temporal dynamics. In contrast, DVS Gesture is less demanding temporally but has greater spatial complexity, for which we adopt both a shallow fully-connected and a deeper convolutional architecture. Details of the experimental setup are provided in Section B of the Appendix, including an ablation study (A.3) to study the importance of normalization, ReLU activation, and having a power-law shaped neuron response.

As noted, online training methods aim to perform temporal credit assignment without relying on BPTT or RTRL. However, depending on how traces or regularizers are formulated to approximate temporal gradients, existing approaches may fail to capture temporal dynamics accurately. To evaluate their impact on accuracy and enable a direct comparison with SpikingGamma SNNs, Table 1 reports our best results alongside those of prior online training methods. We include approaches that either provide results for training with a large number of frames (DVS Gesture) or report any results at all (SHD). For SSC, reliable baselines in the literature were found lacking.

Table 1: Comparison of online training methods for SNNs. SHD results for OTTT, OSTL, and OTPE are taken from (Summe et al., 2024), while the SHD result for DECOLLE is from (Quintana et al., 2024). All other results are taken from the original papers that introduced the respective methods. The architecture is given as [neurons in hidden layers ($\times$ number of hidden layers)] if fully-connected.

| Task | Frames | Method | Architecture | Accuracy ± std |
|------|--------|--------|--------------|----------------|
| DVS Gesture | 500 | DECOLLE | 4-layer CNN, CUBA LIF | 95.54 ± 0.16 |
| | 1000 | SpikingGamma | 4-layer CNN | **96.21 ± 0.31** |
| | 1000 | FPTT | [1024-512], LTC | 91.28 ± 1.05 |
| | 2000 | SpikingGamma | [1024-512] | **93.81 ± 0.18** |
| SHD | 50 | OTTT | [512 × 3], LIF | 71.2 ± 0.8 |
| | 50 | OSTL | [512 × 3], LIF | 70.6 ± 0.7 |
| | 50 | OTPE | [512 × 3], LIF | 75.4 ± 0.5 |
| | 100 | DECOLLE | [450], ALIF | 62.01 ± 0.61 |
| | 100 | ES-D-RTRL | [1024 × 3], RadLIF | 93.35 ± 0.36 |
| | 250 | SpikingGamma | [256 × 3] | **93.55 ± 0.48** |
| SSC | 250 | SpikingGamma | [512 × 3] | **75.91 ± 0.54** |

As shown in Table 1, we find that SpikingGamma SNNs achieve highly competitive test accuracy across all datasets. This is evident both in comparisons with FPTT and DECOLLE on DVS Gesture, and even more so for SHD and SSC, where most other online methods struggle to achieve competitive performance. While DECOLLE performs well on DVS Gesture, it shows limited accuracy on SHD. For SHD, ES-D-RTRL is competitive provided just 100 input frames are used.

**Temporal Precision** Increasing the temporal resolution results in longer sequences, which by itself has a strongly detrimental effect on BPTT performance (Yin et al., 2023). Moreover, BPTT/RTRL

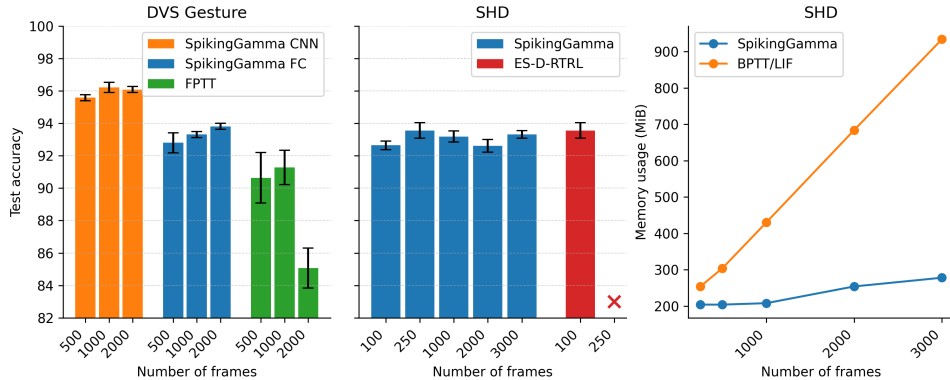

Figure 5: Effect of increasing the number of frames on test accuracy and memory usage. For ES-D-RTRL, the "×" symbol refers to chance level performance. Its performance in between 100 and 250 frames is shown in 6 in the Appendix.

approximations often perform well only after extensive hyperparameter tuning (Xue et al., 2024) and studies of achievable temporal precision are lacking except for comparatively easy datasets such as DVS Gesture (Wang et al., 2024; Yin et al., 2023), which is known to contain limited temporal structure (Chen et al., 2025). Here, we perform a more extensive study by examining how performance is affected at finer temporal resolutions for both DVS Gesture and SHD. To keep the temporal dynamics consistent, we scale the bucket transfer rate by the same factor as the timestep change. To further broaden the comparison, we additionally train with a larger number of frames using publicly available implementations of FPTT and ES-D-RTRL.

In Figure 5, we see that for DVS Gesture, SpikingGamma maintains stable accuracy, even increasing for more frames, while FPTT deteriorates sharply. On SHD, ES-D-RTRL completely collapses, while SpikingGamma maintains the same performance. We can also see that memory usage is low, theoretically constant with time but going up probably due to software overhead, while a BPTT-based method linearly increases with the number of frames.

**Sparsity**   SpikingGamma SNNs can theoretically achieve highly efficient spike-coding, as shown by the coincidence detection task, by keeping most of the dynamics subthreshold and using spike timing codes rather than relying solely on rate codes. This is similar to how delays or adaptive neurons enable efficient neural codes. When applying gain loss on SHD, we reach a spike density comparable to competitive methods (5 to 6 spikes/neuron/sample (Deckers et al., 2024)) with little decrease in accuracy (about 93%), as shown in Figure 8 in the Appendix. However, because the SHD data itself is essentially rate-coded (Cramer et al., 2020), its potential for sparsity is inherently limited (Yu et al., 2025). For datasets with richer and finer temporal structures, we anticipate larger gains in sparsity.

## 5   DISCUSSION AND CONCLUSION

We introduced the SpikingGamma model, a training framework for SNNs that computes exact gradients and thus eliminates the approximation errors that limit existing online methods to learn at high temporal resolutions. Our results show that SpikingGamma SNNs scale to finer temporal resolutions than previously possible, while preserving the ability to capture long-range and complex temporal dependencies. By maintaining information through subthreshold dynamics, it further encourages sparse spike coding, making the resulting models more compatible with the communication constraints of neuromorphic hardware. These advantages bring SNNs closer to large-scale deployment and suggest SpikingGamma models as a foundation for energy-efficient neuromorphic AI.

For proper temporal tasks, even higher levels of sparsity could be achieved by further exploiting SpikingGamma's ability to train complex temporal feature detectors at both the neuron and synapse level. Such tasks, however, are lacking, as most large-scale benchmarks are defined by rate-code models, making it difficult to realize highly efficient temporal codes (Yu et al., 2025). Finally, with modern AI dominated by feedforward network architectures such Transformers, similar modifications to SpikingGamma seem promising to achieve scalable and powerful deep SNNs for sequence learning.

## REPRODUCIBILITY STATEMENT

The methods and appendices should include sufficient details to enable reproduction of the results. To further support reproducibility, a link to the source code is provided in the supplementary material.

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

## A    EXTENDED RESULTS

### A.1    PERFORMANCE COMPARISON WHEN INCREASING NUMBER OF FRAMES

| Frames | SpikingGama (CNN) | SpikingGama (FC) | FPTT |
|--------|-------------------|------------------|------|
| 500 | 95.58 ± 0.18 | 92.80 ± 0.62 | 90.64 ± 1.56 |
| 1000 | 96.21 ± 0.31 | 93.31 ± 0.18 | 91.28 ± 1.05 |
| 2000 | 96.08 ± 0.18 | 93.81 ± 0.18 | 85.07 ± 1.24* |

Table 2: Test accuracy for models trained on DVS Gesture. *Directly reproduced from `https://github.com/byin-cwi/sFPTT`.

| Frames | SpikingGama | ES-D-RTRL |
|--------|-------------|-----------|
| 100 | 92.64 ± 0.27 | 93.35 ± 0.36 |
| 250 | 93.55 ± 0.48 | 4.5* |
| 1000 | 93.18 ± 0.34 | - |
| 2000 | 92.61 ± 0.39 | - |
| 3000 | 93.31 ± 0.24 | - |

Table 3: Test accuracy for models trained on SHD. *Directly reproduced from `https://github.com/chaobrain/brainscale-exp-for-snns`. Number of epochs was reduced from 100 to 30. For 250+ frames, ES-D-RTRL no longer converges, see Figure 6 for transition and collapse from 100 to 300 frames. Therefore, we did not reproduce for 1000 or more frames.

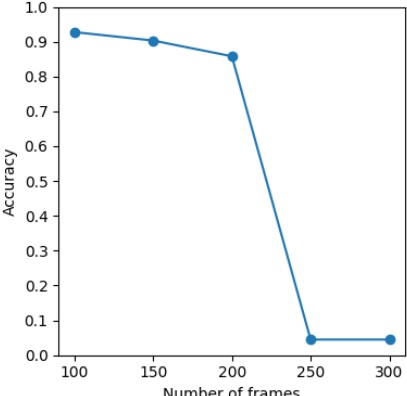

Figure 6: Test accuracy for ES-D-RTRL (reproduced) trained on SHD from 100 to 300 frames. The accuracy for 250 frames was verified over 3 independent training sessions.

## A.2 EFFECT OF INTRODUCING A VARIABLE THRESHOLD

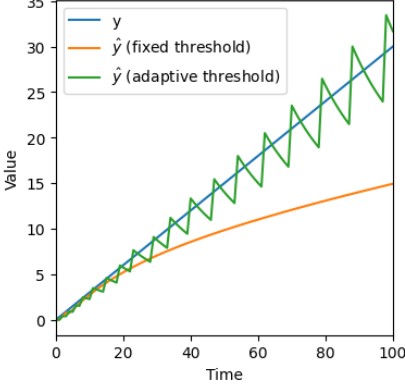

Figure 7: A demonstration showing that, without adaptive thresholding, $\hat{y}$ is unable to accurately estimate higher values of $y$, whereas the use of adaptive thresholding enables $\hat{y}$ to effectively estimate these higher values.

## A.3 ABLATION STUDY ON SHD

Table 4: Ablation study results for SHD

| Ablated | Accuracy ± std |
|---|---|
| No normalization, bucket weights on neuron | 51.95 ± 9.40 |
| No normalization, bucket weights on synapse | 88.81 ± 3.22 |
| RMS normalization (Zhang & Sennrich, 2019) | 93.52 ± 0.42 |
| Batch normalization (Ioffe & Szegedy, 2015) | 28.64 ± 4.31 |
| Batch normalization through time (Kim & Panda, 2021) | 91.24 ± 0.57 |
| No ReLU | 86.20 ± 1.41 |

Table 5: Results from experiments with alternative kernel parameters and shapes. The "Power-law" shape type refers to the default initialization method as introduced in section 3.3, while "Fixed rate" refers to an alternative scheme where $\alpha_k$ is fixed to one value (given as "Rate") for all $K$ kernels.

| Shape type | Rate / $F$ | $K$ | Neuron response | Accuracy |
|---|---|---|---|---|
| Fixed rate | 0.82 | 10 |  | 76.94 ± 0.35 |
| Fixed rate | 0.15 | 10 |  | 68.49 ± 1.45 |
| Power-law | 1.00 | 1 |  | 35.67 ± 1.74 |
| Power-law | 0.00 | 1 |  | 23.62 ± 4.18 |
| Power-law | 0.90 | 10 |  | 85.06 ± 1.07 |
| **Power-law** | **0.15** | **10** |  | **93.55 ± 0.48** |
| Power-law | 0.01 | 10 |  | 87.94 ± 0.57 |

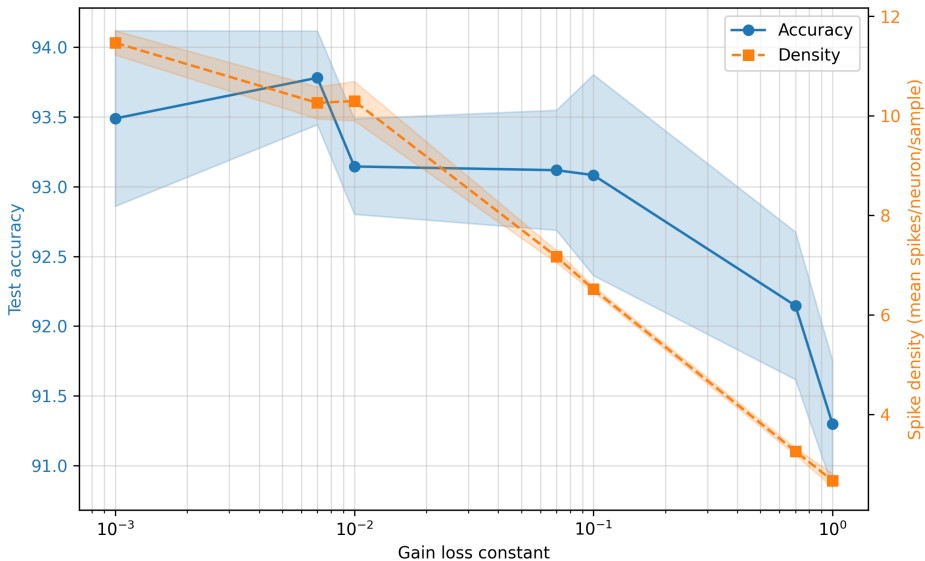

Figure 8: Effect of the gain-loss constant on accuracy and spike density (shown as mean ± std). The spike density is computed using Eq. (16).

## B    DETAILS ON THE EXPERIMENTAL SETUP

### B.1    DATASETS

The Spiking Heidelberg Digits (SHD) and Spiking Speech Commands (SSC) datasets both consist of auditory recordings. Each recording was converted into spike trains using a biologically inspired cochlear model, capturing fine-grained temporal structure in the audio (Cramer et al., 2020). The SHD dataset contains 8156 training and 2264 test samples for 20 spoken digits. The SSC dataset contains 75466 training, 9981 validation, and 20382 test samples for 35 spoken digits. We included the validation samples in the training set (and did no validation during training). For both datasets, the 700-channel cochlear outputs are downsampled to 140 channels to reduce input dimensionality.

The DVS Gesture dataset (Amir et al., 2017) contains event-based visual recordings of 11 hand and arm gestures captured using a DVS camera. It includes 1176 training and 288 test samples. The original $128 \times 128$ event frames are downsampled to $32 \times 32$ by summing up events in a $4 \times 4$ window.

To prepare the event streams from the datasets for model input, the timestamps are discretized into time-bins along the temporal axis. All events falling within the same time-bin are accumulated at their corresponding spatial indices, producing a sequence of frames. These frames are then fed to the model as inputs at successive discrete timesteps.

For computing the mean and standard deviation of the reported accuracies, we repeated the experiments with SHD and SSC 5 times, and with DVS Gesture 3 times.

|  | SHD | SSC | DVS Gesture |
|---|---|---|---|
| Architecture | [256×3] | [512×3] | B.2.1 |
| Timestep size | 3.6 ms | 3.6 ms | 6 ms |
| Number of frames | 250 | 250 | 1000 |
| Batch size | 32 | 32 | 64 |
| Epochs | 30 | 30 | 150 |
| Initial learning rate | 1e-3* | 1e-3* | 1e-3 |
| Learning rate schedule | Step | Step | None |
| Schedule step size | 10 | 10 | |
| Schedule step gamma | 0.1 | 0.1 | |
| Minimum threshold $\vartheta_0$ | 0.2 | 0.2 | 0.2 |
| Number of buckets $K$ | 10 | 7 | 10 |
| Bucket transfer rate $F$ | 0.15 | 0.15 | 0.1 |
| Dropout | 0.1 | 0.1 | 0 |
| Loss function | CE | CE warm-up (B.3.1) | CE |

Table 6: Network architecture and hyperparameters. The architecture is given as [neurons in hidden layers $\times$ number of hidden layers] if fully-connected. Whenever we change the number of frames / the timestep size from the values mentioned above, we scale the bucket transfer rate by the same factor as the timestep size change to keep the temporal dynamics consistent. *Times $K$ per layer upward starting from the output towards the input.

## B.2 NETWORK ARCHITECTURES AND HYPERPARAMETERS

### B.2.1 DVS GESTURE MODEL ARCHITECTURE

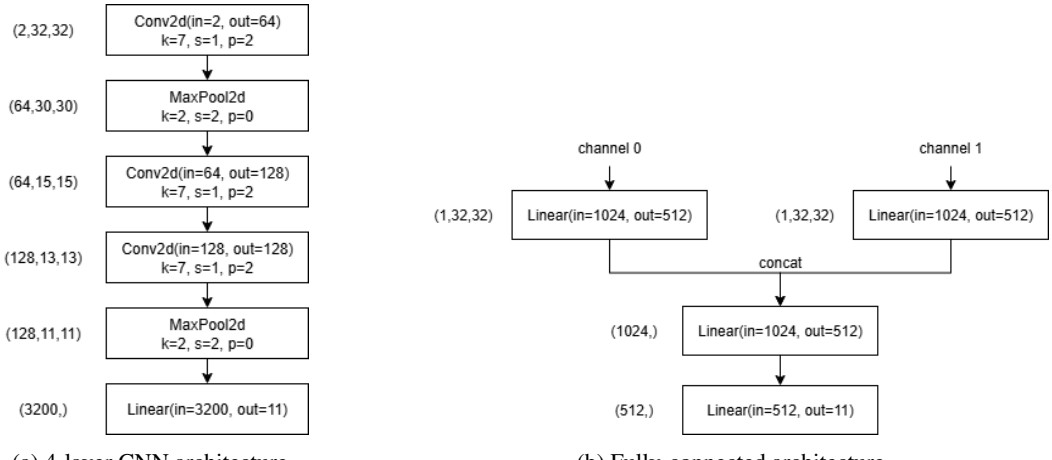

(a) 4-layer CNN architecture  (b) Fully-connected architecture

Figure 9: (a) shows the 4-layer CNN architecture used for DVS Gesture (similar to what was used for DECOLLE in (Kaiser et al., 2020)). (b) shows the fully-connected architecture (similar to what was used for FPTT in (Yin et al., 2023)). In both Figures, neuron layers (that include normalization and dropout if applicable) are omitted; they are placed after each Conv2d or Linear layer.

## B.3 PERFORMANCE METRICS

### B.3.1 CE WARM-UP LOSS

For SSC, there is a delay in sample onset. This can be problematic when training online. To improve performance, we introduce a custom loss function that trains to mute the output at the start of the sample. It interpolates between MSE to zero and CE loss using a time-dependent weight:

$$L(t) = \beta(t) \cdot L^{\text{MSE}}(\hat{y}_{\text{out}}(t), 0) + (1 - \beta(t)) \cdot L^{\text{CE}}(y_{\text{out}}(t), y_{\text{true}}(t)) \tag{14}$$

where

$$\beta(t) = \beta_0 \cdot (\beta_{\text{decay}})^t. \tag{15}$$

with $\beta_{\text{decay}} = 0.99$.

### B.3.2 SPIKE DENSITY

$$\text{spike density} = \frac{1}{N_{\text{samples}} \cdot N_{\text{neurons}}} \sum_{i=1}^{N_{\text{samples}}} N_{\text{spikes}}[i] \tag{16}$$

where $N_{\text{samples}}$ is the dataset size, $N_{\text{neurons}}$ is the total number of neurons in the hidden layers, and $N_{\text{spikes}}[i]$ is the spike count recorded during the inference of the $i$-th sample.

