# OpenReview forum: "Exact Online Learning with Gamma-memory delays for Accurate Feedforward SNNs"
_ICLR.cc/2026/Conference — Submitted to ICLR 2026_

### Official Review · Reviewer_ebL6 · 2025-10-26

**Soundness:** 3
**Presentation:** 3
**Contribution:** 2
**Rating:** 4
**Confidence:** 4

**Summary:**

This paper introduces SpikingGamma, a temporal-coding training framework for SNNs using adaptive recursive memory and sigma-delta spike generation. The method supports direct gradient backpropagation without surrogate gradients, eliminates neuron self-recurrency, enables online learning, and achieves competitive performance on DVS Gesture, SHD, and SSC datasets with significantly improved scalability under high temporal resolution. The authors also demonstrate precise delay learning and coincidence detection tasks to support biological interpretability.

**Strengths:**

1. Temporal coding capability. Demonstrated capability in: precise spike timing learning, delay-based feature detection, biologically plausible mechanisms similar to auditory localization.

3. Scalability to high temporal resolution. Maintains accuracy while other online training methods collapse.

4. Strong experimental results on multiple neuromorphic datasets. Particularly SHD/SSC, where online methods typically underperform.

**Weaknesses:**

1. Novelty positioning is insufficiently clear. The proposed model largely builds on previously known components (Gamma-model, temporal coding, fractional predictive mechanisms), but the exact conceptual innovation versus prior models remains fuzzy.

2. Limited comparison with modern direct-training SNN baselines. Experiments omit:
   * Advanced temporal-coded SNNs.
   * State-of-the-art surrogate gradient-based deep SNNs.

3. Models beyond small FC/CNN architectures remain untested. No exploration of:
   * deeper networks (10+ layers).
   * spiking Transformer/general sequence architectures.

**Questions:**

1. Where is the fundamental theoretical novelty compared to Fractionally Predictive SNNs (Bohte 2011) or Gamma models (De Vries 1992)?

2. How does the model behave without adaptive thresholding? This is a core contribution in paper [3], which is also about temporally coded SNNs.

3. The proposed method is only validated on event-driven benchmarks. It remains unclear whether the model can be applied to static datasets (e.g., CIFAR-10, ImageNet).

4. Missing comparisons with state-of-the-art temporal-coding SNNs[1~3]. Although the paper emphasizes precise spike-timing learning capabilities, the comparisons mainly involve online training or BPTT approximation methods, not true timing-based SNN baselines.

[1] Training spiking neural networks with event-driven backpropagation. NeurIPS 2022.

[2] Exploring loss functions for time-based training strategy in spiking neural networks. NeurIPS 2023.

[3] Temporal spike sequence learning via backpropagation for deep spiking neural networks. NeurIPS 2020.

---

> ### Author Response · Authors · 2025-11-19
>
> We would like to thank the reviewer for taking the time to read our paper and for acknowledging the strengths of the work. We address the concerns in order.
>
> The first point concerns novelty (and answers question 1). Indeed, we took inspiration from earlier ideas, but both main lines of prior work faced significant scaling difficulties. In the Gamma model by De Vries, the reliance on BPTT made the approach hard to scale. In the fractional predictive coding work by Bohte, the kernels did not scale, and the computation was performed per synapse, which quickly became expensive.
>
> Our work overcomes these issues by introducing a scalable kernel representation based on buckets and combining this with the fractional predictive coding work. This structure is inspired by the Gamma model but differs in essential points. The Gamma model used learnable weights between kernels, while in our model these weights, the bucket transfer rates, are fixed. This is crucial for not needing BPTT, which the original Gamma model still requires. We also introduce trainable bucket weights at the neuron level, which makes the model far cheaper to train while still functioning correctly when combined with proper normalization. We further show the need for adaptive thresholding to maintain the dy_hat over dy equals one rule described in Section 3.3. Finally, we combine these components with the training of synaptic weights. Without these novelties, the earlier works simply do not scale to real benchmark problems.
>
> Regarding the comparisons, we intentionally focused on the specific context in which we claim an advantage. To this end, we restricted comparisons to methods that (1) report results on complex temporal datasets rather than static datasets such as CIFAR-10 or their neuromorphic variants such as CIFAR-10-DVS, and that (2) have shown some capacity to operate at fine temporal resolutions. The first criterion excludes most temporal-coding methods, including the works [1-3] suggested by the reviewer in question 4. These methods are evaluated on mostly static tasks, and it remains unclear whether they can learn complex temporal structure in realistic temporal classification tasks. The second criterion excludes all BPTT-based approaches, including state-of-the-art surrogate-gradient SNNs, because BPTT does not scale well in the temporal dimension. These models are known to break at high temporal resolutions, while our method remains stable. This is why we did not include them. We also considered including EventProp as an additional comparison, since it performs reasonably on SHD and SSC and uses less GPU memory than BPTT, but because its spatial scalability is questionable and its implementation is complex to use, we decided not to include it. If the reviewer believes this comparison should be added, we are happy to do so.
>
> We agree that advanced temporal-coded SNNs could be interesting to explore. Our coincidence-detection experiment was intended to illustrate the potential in this direction and to reflect earlier results from [Bohte, 2011]. The timing-based works [1-3] provide useful intuition. However, training a deep network for precise temporal coding on complex temporal tasks with an online update rule remains largely open research. We believe this deserves a separate project of its own, which is why we did not make strong claims in this direction in the present paper.
>
> The reviewer also raises concerns about the lack of experiments on larger architectures. For the datasets we considered, larger models do not offer much benefit. For larger datasets with more complex spatial variations, deeper CNNs or vision transformers would indeed be needed, and extensive research already exists on how SNNs can manage spatial complexity with initialization schemes, normalization strategies, dropout, and other mechanisms. Our work does not make claims in this spatial direction. Our focus is a different kind of scaling: scaling in the temporal dimension with theoretically constant memory while still learning complex temporal dependencies at the neuron level. As the reviewer notes, this is a difficulty for online SNN training, and our results demonstrate scaling beyond any prior work, up to 3000 timesteps without performance degradation.
>
> A remaining question may be whether our temporal mechanism interferes with spatial learning. This is why we included DVS Gesture. Although the dataset offers limited temporal complexity, it requires CNNs for competitive performance. Our method shows no degradation relative to existing approaches. In our view, further experiments on larger spatial datasets would mostly confirm the absence of negative interactions rather than strengthen the temporal claims that form the focus of our work.

---

> > ### Author Response · Authors · 2025-11-19
> >
> > Finally, we answer the reviewer’s remaining questions 2 and 3. Without adaptive thresholding, the model performs poorly. Adaptive thresholding is required to maintain the dy hat over dy equals 1 rule described in Section 3.3 and illustrated in Section A.2 in the Appendix. Concerning static datasets, our method can in principle be applied to them, but static datasets are less aligned with the temporal-scaling focus of the paper and offer little insight into the claims we make.
> >
> > We thank the reviewer again for the constructive comments. We hope this clarification helps communicate the novelty, scope, and strengths of our work more clearly.

---

### Official Review · Reviewer_hj14 · 2025-10-31

**Soundness:** 2
**Presentation:** 2
**Contribution:** 1
**Rating:** 2
**Confidence:** 5

**Summary:**

The authors of this paper present a new approach/or spiking model called SpikingGamma, which they claim is able to produce better results than the standard spiking models trained with BPTT.

**Strengths:**

**Pros**:

- Looking at new spiking models is an interesting direction, and if successful, can yield improved results.
- The paper is easy to follow.

**Weaknesses:**

**Cons:**

- In the abstract and the main body, the authors position their work as an alternative approach to BPTT with surrogate gradients. This is simply a misstatement since the proposed SpikingGamma is a modified spiking model as opposed to a new training method.

- SpikingGamma uses a ReLu activation - this is why surrogate gradients are not necessary; however, this also means the model is less biologically plausible.

- SpikingGamma integrates ideas from several earlier works: the Gamma-model (De Vries & Principe, 1992), the Temporal Kernel RNN (Sutskever & Hinton, 2010) and the Fractionally Predictive SNN (Bohte, 2011; Rombouts & Bohte, 2010). Not clear what new innovations this work presents.

- The proposed model is a lot more complex than the standard LIF model; this complexity needs to be justified.

- While being motivated by the earlier Gamma-model, the authors don't motivate the adoption of sigma-delta spike coding and provide intuition about why this improves the performance at the network level.

- The experimental validation is inadequate; only very small datasets have been used (no Imagenet level benchmarks). The demonstrated improvement is not significant, particularly given the significant amount of increase in the model complexity.

**Questions:**

- In (2) and elsewhere, does superscript k indicate time step?

- In (5), how is $\hat{y}^k$ initialized at the first time step (t = 0)?

---

> ### Author Response · Authors · 2025-11-19
>
> We agree with the reviewer that exploring new spiking neuron models can yield new insights. The LIF neuron, probably the most well-known model, has clear limitations. In particular, the plain LIF neuron is largely incapable of learning complex temporal features. This shows up already on a dataset like SHD, where it is difficult to exceed about 85 percent test accuracy. Many extensions to the LIF have been proposed that improve temporal learning, and our work similarly introduces a neuron model that achieves better performance on temporally challenging datasets such as SHD and SSC.
>
> However, the reviewer seems to interpret our contribution as stopping at the introduction of a new neuron model, while this is only one component of the approach. The key idea is that the neuron model is coupled to a mechanism that maintains input history through the bucket structure, which effectively provides a rolling temporal representation. This is what allows the network to access and learn from (long) temporal dependencies without relying on BPTT. This is also why we describe the method as an alternative to BPTT. When BPTT is removed in existing models that rely on it for learning temporal structure, their performance collapses. In contrast, our model continues to learn effectively because the relevant temporal information is preserved within the buckets.
>
> The reviewer raises concerns about neuron complexity. In practice, the model is not as complex as it may seem. As with a LIF neuron, there are synaptic weights, and in addition there is a small set of bucket weights per neuron (a characteristic of the per-neuron filter model that was used for all benchmarks). The number of extra parameters therefore scales linearly with the number of neurons, similar to how for example the number of parameters of layer norm scales. The bucket dynamics replace the single exponential decay factor of a LIF neuron with K decay factors. These operations are performed per neuron and can be computed in parallel, so the computational overhead is limited. Once BPTT is removed, the overall computational efficiency is actually high.
>
> The reviewer also questions the novelties of the work. Indeed, we took inspiration from earlier ideas, but both main lines of prior work faced significant scaling difficulties. In the Gamma model by De Vries, the reliance on BPTT made the approach hard to scale. In the fractional predictive coding work by Bohte, the kernels did not scale, and the computation was performed per synapse, which quickly became expensive.
>
> Our work overcomes these issues by introducing a scalable kernel representation based on buckets and combining this with the fractional predictive coding work. This structure is inspired by the Gamma model but differs in essential points. The Gamma model used learnable weights between kernels, while in our model these weights, the bucket transfer rates, are fixed. This is crucial for not needing BPTT, which the original Gamma model still requires. We also introduce trainable bucket weights at the neuron level, which makes the model far cheaper to train while still functioning correctly when combined with proper normalization. We further show the need for adaptive thresholding to maintain the dy_hat over dy equals one rule described in Section 3.3. Finally, we combine these components with the training of synaptic weights. Without these novelties, the earlier works simply do not scale to real benchmark problems.

---

> ### Author Response · Authors · 2025-11-19
>
> The reviewer also remarks that the experimental validation is limited and that the improvements are of limited significance. We want to clarify that without our method, none of the existing online methods can achieve strong performance on SHD or SSC at fine temporal resolution. They break down entirely. In contrast, our model reaches up to 93 percent accuracy under these conditions. We consider the difference between unstable performance below 80 percent and stable high performance above 90 percent to be substantial. To our knowledge, we are currently the only method that demonstrates good results in this online training regime with many timesteps.
>
> Regarding the absence of larger datasets such as ImageNet, we want to emphasize that such datasets involve complex spatial structures that require deep CNNs. A large body of work exists on spatial scaling of SNNs, but our contribution is not related to spatial scaling, and we do not claim improvements in that direction. Our focus is on temporal scaling with constant memory, together with the ability to learn complex temporal features at the neuron level. Datasets such as SHD and SSC were chosen precisely because they stress this temporal aspect. In this setting our experimental support is strong, and we demonstrate stable performance up to 3000 timesteps without degradation, which goes well beyond what has previously been shown.
>
> A remaining question may be whether our temporal mechanism interferes with spatial learning. This is why we included DVS Gesture. Although the dataset offers limited temporal complexity, it requires CNNs for competitive performance. Our method shows no degradation relative to existing approaches. In our view, further experiments on larger spatial datasets would mostly confirm the absence of negative interactions rather than strengthen the temporal claims that form the focus of our work.
>
> Regarding the adoption of sigma-delta spike coding, we agree that this deserves more explicit motivation. The sigma-delta mechanism ensures that only changes in the predictive signal trigger spikes, which keeps the activity sparse and makes the representation robust under high temporal resolutions. This aligns naturally with the rolling temporal representation produced by the buckets. Furthermore, it allows us to backpropagate from y_hat (computed through sigma-delta coding), directly to y, avoiding the need to backpropagate through spikes. Much more theoretical background is given in [1]. We will make this motivation clearer in the revised manuscript.
>
> We now address the reviewer’s specific questions. In equation (2) and elsewhere, the superscript k denotes the kernel index, as defined in rule 151 below equation (2). Equation (5) gives the update rule. At time zero all kernels are initialized to zero. This will be clarified in a revision.
>
> We hope that this clarifies the contribution of our work and its relevance. The ability to train complex temporal features across arbitrarily long timescales without relying on BPTT is, to our knowledge, unique to our method and of significant importance for future neuromorphic applications.
>
> [1] Rombouts, J., & Bohte, S. (2010). Fractionally predictive spiking neurons. Advances in Neural Information Processing Systems, 23.

---

### Official Review · Reviewer_YeHk · 2025-11-01

**Soundness:** 2
**Presentation:** 3
**Contribution:** 2
**Rating:** 6
**Confidence:** 2

**Summary:**

This paper discusses the issue of approximate gradients when training SNNs using STBP, and introduces the Gamma model to circumvent approximate gradients during training. The effectiveness of this method has been verified on some small datasets and network architectures.

**Strengths:**

This paper exhibits robust logical reasoning, featuring a meticulously organized overview of current methods and an introduction to this paper's motivation. The schematic representations of the methods are visually appealing, and the description of the technique is exhaustive. However, given my limited expertise in this domain, I am challenged to evaluate the novelty of the method.

**Weaknesses:**

The method presented in this paper is comprehensive. However, I have several inquiries regarding the experimental section.

1. Dataset: The experiments documented in this study predominantly utilize the DVS dataset. It would be beneficial to ascertain whether the efficacy of SpikingGamma is sustained when applied to more common datasets, such as RGB video data.

2. Network Architecture: An observation from Table 1 indicates that the majority of the networks are characterized by a reduced structure, specifically comprising four layers or fewer. A pertinent question arises as to whether SpikingGamma can be effectively adapted for use in networks with an increased number of parameters, such as those that are deeper and wider.

**Questions:**

I notice that this paper mentions "to reduce the memory footprint of BPTT" in lines 93-94. Does "memory" here refer to the GPU memory required for training? I also notice that OTTT and its successor SLTT [1], as well as some network architecture design-related work, T-RevSNN [2], both can save GPU memory consumed during BPTT training. Could you discuss these methods in more detail?

[1] Towards Memory- and Time-Efficient Backpropagation for Training Spiking Neural Networks. ICCV 2023.

[2] High-Performance Temporal Reversible Spiking Neural Networks with $O(L)$ Training Memory and $O(1)$ Inference Cost, ICML 2024.

---

> ### Author Response · Authors · 2025-11-19
>
> We would like to thank the reviewer for the positive review. Below we address the weaknesses and then respond to the questions.
>
> Regarding the datasets, we indeed chose neuromorphic and DVS-style benchmarks. These datasets capture the sparse, event-based data that SNNs are particularly well suited to handle, and they contain rich temporal structure that is highly relevant to real-world neuromorphic applications. We selected SHD and SSC in particular because they are widely used, have complex temporal dependencies, and allow us to study precisely the kind of temporal scaling that our work targets. The reviewer mentions more common datasets such as RGB video data. Although such datasets do have a temporal component, most of the complexity resides in the spatial domain. Since RGB data is dense, the typical advantages of SNNs are less pronounced, and exploring how SNNs should be adapted for those settings is outside the scope of the present work. Our focus lies in scaling along the temporal dimension as an alternative to BPTT, and for this purpose we believe our chosen datasets are well aligned with the claims we make.
>
> The concern regarding network size is closely related. We indeed use relatively small networks, but these architectures are appropriate for the datasets in question and consistent with prior work. For datasets where spatial complexity dominates, deeper or wider networks would be necessary, but that line of research is not what we claim to advance. By including DVS Gesture we did verify that SpikingGamma can also operate in multilayer CNN settings and achieves performance comparable to other methods. For the spatial aspect, we think this provides sufficient evidence that our method does not introduce limitations relative to existing approaches.
>
> Methods such as OTTT reduce memory by maintaining and updating eligibility traces that approximate BPTT or RTRL. SLTT removes the temporal gradients entirely and keeps only the spatial gradients, which avoids storing temporal activations. While these approaches are effective in certain regimes, they have not been shown to learn complex temporal features or to remain stable over many timesteps. T-RevSNN uses reversible architectures to avoid storing activations, but it similarly does not address long timescale temporal dependencies.
>
> Our approach follows a different principle altogether. Instead of maintaining traces through time, SpikingGamma uses a rolling temporal mechanism that resembles a learned temporal convolution. This allows us to learn complex temporal dynamics directly, without unrolling through time and without tracking per-timestep states. As a result, our memory usage remains constant in theory and our accuracy remains stable even when training over thousands of timesteps.
>
> We hope this clarifies how our choice of datasets and architectures relates to the goals of the paper and how our approach fundamentally differs from OTTT, SLTT and T-RevSNN with respect to temporal learning.

---

### Official Review · Reviewer_gExY · 2025-11-08

**Soundness:** 3
**Presentation:** 2
**Contribution:** 3
**Rating:** 4
**Confidence:** 4

**Summary:**

This paper addresses the significant challenge of training Spiking Neural Networks (SNNs), particularly the poor scaling of current methods when using the fine temporal discretization required for neuromorphic hardware. State-of-the-art approaches, which treat SNNs as RNNs trained with BPTT/RTRL and Surrogate Gradients (SGs), scale poorly with temporal resolution and can be unstable

**Strengths:**

The paper directly addresses the inability of current SNN training methods to scale to high temporal resolutions. The most significant result is in Figure 5, which shows that SpikingGamma's performance is stable as the number of timesteps increases, whereas other online methods like FPTT and ES-D-RTRL see their accuracy "deteriorate sharply" or "completely collapse". This robustness is a critical property for real-world neuromorphic applications

The model is feedforward  and does not require unrolling through time for gradient computation (like BPTT). Figure 5 (right) shows that its memory usage is "theoretically constant" with time, a stark contrast to the linear memory growth of BPTT. This makes the method genuinely suitable for online processing and resource-constrained hardware.

**Weaknesses:**

The experimental evaluation is confined to datasets (SHD, SSC, DVS Gesture) that are relatively small in scale. The paper does not provide evidence of how the SpikingGamma model's performance, stability, and training efficiency would scale to much larger and more complex event-based benchmarks, such as N-ImageNet or UCF-DVS.

The paper's claim of an "online" and "constant memory" learning mechanism (a key advantage over BPTT) requires clarification. The method computes a gradient at each timestep $t$ based on a loss $L(t)$ available at that same step. This is effective for the chosen benchmarks, which allow for dense or accumulated loss signals.

However, it is unclear how this framework applies to tasks with sparse, delayed rewards or a single classification label available only at the end of a long sequence (e.g., $T=3000$). These scenarios are a primary motivation for using SNNs.

 If the loss is only available at the final timestep $T$, does the model need to store the intermediate "bucket" states ($\hat{y}_i^k$) from all preceding timesteps to compute the gradients?

 If so, this would seem to negate the "constant memory" advantage, as it would reintroduce a memory cost that scales linearly with sequence length, similar to BPTT.

 Alternatively, if gradients are computed at each step, accumulated, and applied at the end, how does this fit the definition of "online" learning? This ambiguity regarding the model's update rule and memory footprint in sparse-reward settings is a significant weakness.

**Questions:**

see weakness below

---

> ### Author Response · Authors · 2025-11-19
>
> We would like to thank the reviewer for acknowledging the strengths of our work, in particular the importance of stability at high temporal resolutions and its relevance for neuromorphic applications. The reviewer’s first concern relates to the use of relatively small datasets and whether our approach can scale further. To address this, it is important to clarify the type of scaling that is relevant for our contribution.
>
> The reviewer mentions datasets such as N-ImageNet or UCF-DVS, which indeed involve much richer spatial complexity. Many works in SNNs and ANNs explore spatial scaling through deeper CNNs, vision transformers, normalization strategies, improved initializations and so on. However, our work does not target improvements in spatial feature learning. Our focus is on a different kind of scaling: the ability to handle very large numbers of timesteps with theoretically constant memory, while still being able to learn complex temporal responses at the neuron level. As the reviewer has noted, this is a very important aspect for neuromorphic systems. In this context, our experiments show scaling further than any prior work to our knowledge, up to 3000 timesteps, and importantly without performance degradation. We believe the evidence supporting these specific claims is solid.
>
> A remaining question may be whether our temporal mechanism interferes with spatial learning. This is why we included DVS Gesture. Although the dataset offers limited temporal complexity, it requires CNNs for competitive performance. Our method shows no degradation relative to existing approaches. In our view, further experiments on larger spatial datasets would mostly confirm the absence of negative interactions rather than strengthen the temporal claims that form the focus of our work.
>
> The reviewer also raises the question of sparse or delayed rewards, which is indeed an interesting scenario. For the tasks we experimented with, we compute gradients at each step. This is online in the sense that we only keep the activations for the current timestep, compute the loss, accumulate gradients and discard everything except the accumulated gradients. Nothing else needs to be stored, tracked or backpropagated through time.
>
> If a label would only be available at the final timestep, the model can still learn from the information that is present in the buckets at that moment. If the buckets empty too quickly in such a setting, one can slow down the decay or introduce more buckets so that relevant information persists until the end. This does not negate the constant memory advantage, although it may lead to slower learning in tasks with extremely delayed supervision. In general, this is a challenge faced by any learning method, and the choices of what to keep, how long to keep it and what to discard inevitably depend on the task.
>
> We hope these clarifications help contextualize the scope of our contribution and the settings in which SpikingGamma offers clear benefits.

---

### Meta-Review · Area_Chair_pFth · 2025-12-31

**Summary:**

The paper proposes "SpikingGamma," a model designed to enable exact online training of feedforward Spiking Neural Networks (SNNs) by utilizing recursive memory structures to handle fine temporal resolutions without the instability of surrogate gradients or the memory costs of BPTT. The reviewers generally acknowledged the paper's core strength: the ability to maintain stability and accuracy at high temporal resolutions where other online methods (like FPTT and ES-D-RTRL) fail. The theoretical constant memory usage was also highlighted as a significant advantage for neuromorphic hardware implementation.

However, the consensus decision leans toward rejection or weak acceptance due to the experimental scope. A primary concern shared by nearly all reviewers (gExY, hj14, ebL6) was the reliance on relatively small, event-driven datasets (SHD, SSC, DVS Gesture). Reviewers requested validation on larger-scale, spatially complex benchmarks like N-ImageNet or RGB video to prove the method's generality. Additionally, there were significant questions regarding the novelty of the method compared to older Gamma-memory and fractional predictive coding models , as well as the validity of "online" learning claims in sparse-reward scenarios.

**Reviewer Concerns:**

### **Addressed Concerns**

**Definition of "Online" Learning**: Reviewer gExY questioned how the method handles sparse rewards (e.g., a label only at $T=3000$) while maintaining constant memory. The authors successfully clarified that they compute gradients at each step, accumulate them, and discard activations, meaning they do not need to unroll through time or store history buffers even if the reward is delayed.

**Novelty regarding Gamma/Fractional Models**: Reviewers hj14 and ebL6 questioned the innovation over De Vries (1992) and Bohte (2011). The rebuttal effectively distinguished the proposed method by explaining that original Gamma models required BPTT due to learnable rates (whereas SpikingGamma uses fixed rates and trainable neuron weights), and Bohte’s method failed to scale due to per-synapse computation.

Comparison to OTTT/SLTT: Reviewer YeHk asked for comparisons to other memory-efficient methods like OTTT and SLTT. The authors addressed this by explaining that OTTT uses approximations and SLTT drops temporal gradients entirely, whereas SpikingGamma captures precise temporal dependencies via the bucket mechanism.

### **Remaining Concerns**
**Scalability to Complex Spatial Tasks**: The authors argued that scaling to datasets like ImageNet is a spatial challenge, whereas their contribution is temporal. While logically consistent, this is unlikely to satisfy reviewers (gExY, hj14) who regard performance on large-scale standard benchmarks as a prerequisite for acceptance at top-tier conferences.

**Limited Baselines**: Reviewer ebL6 requested comparisons to specific state-of-the-art temporal-coding SNNs. The authors excluded these because they operate on static datasets. While the authors' justification is valid, the lack of direct comparison to the strongest current baselines in the field remains an empirical gap.

**Model vs. Training Method**: Reviewer hj14 insisted that the paper misrepresents itself as a training method when it is actually a model modification. The authors argue the model modification is what allows the training method, but this semantic disagreement may still negatively influence the reviewer's view of the paper's framing.

**Reviewer Scores:**

While the rebuttals successfully clarified the technical "online" mechanism for Reviewer gExY and distinguished the method from memory-efficient baselines for Reviewer YeHk, potentially nudging their assessments toward a weak acceptance (scores of 5-6), this is counterbalanced by the persistent skepticism of Reviewers hj14 and ebL6. Reviewer hj14 is likely to retain a low score (2-3) due to fundamental disagreements about the "model vs. training method" framing and biological plausibility, while Reviewer ebL6’s specific request for modern temporal-coding baselines was declined by the authors, limiting their likely score to a borderline 4 or 5.

**Conclusion**: The paper effectively proved its stability claims but failed to satisfy the demand for broader experimental validation (e.g., ImageNet, modern baselines), resulting in a stalemate that leans toward rejection due to the lack of a strong champion willing to overlook the limited experimental scope.

---

### Decision · Program_Chairs · 2026-01-26

Reject